# Systematic review and meta-analysis of pregnancy outcomes in women with polycystic ovary syndrome

Mahnaz Bahri Khomami [1] ✉, Soulmaz Shorakae[1], Somayeh Hashemi[2], Cheryce L. Harrison[1,3], Terhi T. Piltonen [4], Daniela Romualdi [5], Chau Thien Tay[1], Helena J. Teede [1,3], Eszter Vanky[6,7,8] & Aya Mousa[1,8]

Screening for polycystic ovary syndrome (PCOS) in antenatal care is inadequate, largely owing to the lack of clarity around whether PCOS is an independent risk factor for pregnancy complications. This systematic review and meta-analysis include 104 studies and 106,690 pregnancies in women with and without PCOS from inception until 13th July 2022. We report that women with PCOS are younger and have higher body mass index (BMI) around conception and have greater gestational weight gain. The odds of miscarriage, gestational diabetes mellitus, gestational hypertension, pre-eclampsia and cesarean section are higher in women with PCOS. The increased odds of adverse outcomes in PCOS remain significant when age and BMI are matched and when analyses are restricted to high-quality studies. This work informed the recommendations from the 2023 international evidence-based guideline for the assessment and management of polycystic ovary syndrome, emphasizing that PCOS status should be captured in all women who are planning to, or have recently become pregnant to facilitate prevention of adverse outcomes and improve pregnancy outcomes.

Polycystic ovary syndrome (PCOS) is a lifelong condition and, with a prevalence of 5% to 13%, is the most prevalent endocrinopathy in women of reproductive age[1]. In adults, diagnosis is based on a minimum of two criteria of hyperandrogenism, oligo-anovulation and/or polycystic ovaries[2]. Obesity is common in PCOS[3] and exacerbates intrinsic insulin resistance and the clinical manifestations of PCOS[4]. Anxiety, depression and poor quality of life are also more common in women with PCOS compared with women without PCOS[5]. Affected women are also more likely to have anovulatory infertility[6], requiring treatments such as assisted reproductive technology (ART) to

conceive. While each of the metabolic, psychological, and reproductive features of PCOS is recognized in antenatal care guidelines as an independent risk factor for pregnancy complications, women with PCOS are not universally considered a high-risk group for pregnancy complications[7].

In previous meta-analyses, women with PCOS were shown to have elevated odds of pregnancy complications. These include significantly higher odds of miscarriage, gestational diabetes mellitus (GDM), gestational hypertension, pre-eclampsia, cesarean section[8–12], gestational weight gain[8], and induction of labor[8], compared with women

[1]Monash Centre for Health Research and Implementation, Faculty of Medicine, Nursing and Health Sciences, Monash University, Melbourne, Australia. [2]Brock University, Ontario, Canada. [3]Endocrinology and Diabetes Units, Monash Health, Melbourne, VIC, Australia. [4]Department of Obstetrics and Gynecology, Research Unit of Clinical Medicine, Medical Research Center, Oulu University Hospital, University of Oulu, Oulu, Finland. [5]Department of Woman and Child Health and Public Health, Woman Health Area, Fondazione Policlinico Universitario A. Gemelli, Rome, Italy. [6]Department of Clinical and Molecular Medicine, Faculty of Medicine and Health Sciences, Norwegian University of Science and Technology, Trondheim, Norway. [7]Department of Obstetrics and Gynecology, St. Olav's Hospital, Trondheim University Hospital, Trondheim, Norway. [8]These authors contributed equally: Eszter Vanky, Aya Mousa. ✉ e-mail: mahnaz.bahrikhomami@monash.edu

without PCOS. Instrumental delivery was similar in women with and without PCOS[9,10], while perinatal depression was an untapped area of research in this population[8]. Importantly, the increased odds of pregnancy complications in PCOS were influenced by PCOS diagnostic criteria, age, obesity, conception mode[8], and quality of studies[8,9,11].

While empirical evidence demonstrates that women with PCOS have higher odds of pregnancy complications, substantial clinical heterogeneity was reported in pooled analyses. Furthermore, subgroups and univariate or multivariate meta-regression analyses had small numbers of studies, which were insufficient to obtain meaningful results[8]. In the context of the 2023 update of the International Evidence-Based Guideline for the Assessment and Management of Polycystic Ovary Syndrome[2], we aimed to update our prior systematic review, meta-analysis, and meta-regression[8] to determine the prevalence of pregnancy complications in women with and without PCOS. With an expanded data set, we also aimed to explore whether pregnancy outcomes in women with and without PCOS are affected by age, body mass index (BMI), conception with ART, or high-quality study design.

## Results
The literature search identified 4595 articles published since 2017, of which 39 studies were included in the meta-analysis after full-text review. Agreements between the reviewers in the title and abstract screening and full-text screening were considered good and excellent, with kappa values of 0.72 and 0.94, respectively[13]. There were 77 studies in the 2017 meta-analysis, of which 65 studies were included (after removing those not meeting the updated eligibility criteria for PCOS diagnosis). Combining the new and previous articles, a total of 104 articles were included in the present systematic review (Fig. 1).

The outcomes were reported in 17384 women with PCOS and 89,306 women without PCOS (Supplementary Table 1). Fifty-two studies were conducted in Asia, 26 in Europe, 21 in America, two in Australia and New Zealand and two in Africa. One study recruited women with multiple pregnancies[14]. Six studies reported outcomes in women who took metformin after conception[15–20], and one study reported outcomes in women who had conceived after bariatric surgery[21]. Thirty-three studies reported outcomes in post-ART pregnancies[22–54], and four reported outcomes in pregnancies with GDM in women with and without PCOS[55–58]. Sixteen studies had a high-quality design[18,25,36,48,56,59–69]. Twenty-seven studies matched women with and without PCOS for age[17–19,22,32,36,48,56,59–64,66–78] and 14 for age and BMI[36,48,56,60–64,66,68,70,75,77,78]. No publication bias was found for any of the outcomes. The certainty of evidence for the outcomes was very low to moderate, mainly due to a high risk of bias, serious inconsistency, and serious indirectness. Further details are delineated in the Technical Report for the 2023 International Evidence-based Guideline for the Assessment and Management of Polycystic Ovary Syndrome[79].

Overall, 69 studies reported age, and 58 studies reported BMI either at preconception or at early pregnancy. Women with PCOS were younger (MD: −0.63 years; 95% CI: −0.87, −0.39) and had a higher BMI (1.76 kg/m$^2$; 1.44, 2.08) compared with women without PCOS. Sensitivity analysis showed that after exclusion of studies in which women were taking metformin after conception[16,17,20] or conceived after bariatric surgery[21], lower age (−0.68 years; −0.93, −0.44) and higher BMI (1.68 kg/m$^2$; 1.35, 2.00) remained significant in PCOS. Table 1 shows the effect sizes for outcomes of interest on pooled and sensitivity analyses. Forest and cumulative plots, funnel plots (Supplementary Figs. 1a to 10a) and Egger's test results are provided in the Supplementary information.

### Gestational weight gain
Sixteen studies reported mean gestational weight gain in 755 women with PCOS and 1765 women without PCOS[41,56,58,59,64,71,74,75,77,80–86]. Women with PCOS had higher gestational weight gain compared with women without PCOS (MD: 0.96 kg; 95% CI: 0.01, 1.90). There were no

studies in which women were taking metformin after conception. Cumulative meta-analysis suggested that the MD of gestational weight gain in women with and without PCOS did not substantially change over time; however, studies published before 1999 (n = 2) contributed to heterogeneity. Higher gestational weight gain was not retained in post-ART pregnancies[41], prospective[56,58,75,83–85] and high-quality studies[56,59,64]. Two studies reported gestational weight gain in pregnancies with GDM[56,58]; gestational weight gain was higher in women with PCOS (1.81 kg; 0.42, 3.19). In seven studies, women with and without PCOS were matched for age[56,59,64,71,74,75,77] and in four for both age and BMI[56,64,75,77]; PCOS was associated with gestational weight gain in neither age-matched nor age- and BMI-matched studies.

### Miscarriage
Forty-five studies reported miscarriage[20–31,34–39,42–52,54,62,63,65,66,77,78,87–95]; one study was excluded from the meta-analysis because of overlapping participants[62], leaving 9893 women with PCOS and 57411 women without PCOS. Miscarriage was defined as pregnancy loss within the first six to eight weeks of pregnancy (using serial ultrasound) in one study[34] or within the first 12 weeks of pregnancy in seven studies[36,38,40,54,88,91,93], birth prior to 20 weeks of pregnancy in two studies[38,77], spontaneous pregnancy loss prior to 20 weeks of pregnancy in three studies[24,25,43], prior to 24 weeks of pregnancy in two studies[42,48] or prior to 28 weeks of pregnancy in one study[53] and pregnancy loss encompassing miscarriage and ectopic pregnancy in one study[44]. The remaining 27 studies did not provide a definition. The odds of miscarriage were higher in women with PCOS compared with women without PCOS (OR: 1.49; 95% CI: 1.20, 1.85). Sensitivity analysis showed that after exclusion of studies in which women were taking metformin after conception[20] or conceived after bariatric surgery[21], miscarriage remained higher in women with PCOS (1.53; 1.23, 1.92). Cumulative meta-analysis suggested that the odds of miscarriage in women with and without PCOS did not substantially change over time. The higher odds of miscarriage were retained in post-ART pregnancies[22–29,31,34–40,42–52,54], prospective[22–24,27,31,34,35,37–39,43,45,62,63,65,66,78,89,94] and high-quality studies[25,36,48,63,65,66]. In seven studies, women with and without PCOS were matched for age[22,36,48,63,66,77,78], and in six, for both age and BMI[36,48,63,66,77,78]; PCOS was associated with increased odds of miscarriage in age-matched or age- and BMI-matched studies.

### Gestational diabetes
Fifty-eight studies reported GDM in 10011 women with PCOS and 57041 women without PCOS[14–21,25,32,33,41,42,44,49,53,59,62,63,65,66,68–78,80–83,88,92,94–113]. Three studies were excluded from the meta-analysis because of overlapping participants[15,62,109]. Inconsistent definitions were provided for GDM across the studies. The odds of GDM were higher in women with PCOS compared with women without PCOS (OR: 2.41; 95% CI: 1.95, 2.99). Sensitivity analysis showed that after exclusion of studies in which women were taking metformin after conception[16–20] or conceived after bariatric surgery[21], the odds of GDM remained higher in PCOS (2.46; 1.99, 3.04). Cumulative meta-analysis suggested that the odds of GDM in women with and without PCOS did not substantially change over time. The higher odds of GDM were retained in post-ART pregnancies[25,32,33,41,42,44,49,53,95], prospective[63,65,66,70,73,75,76,78,83,88,94,100,102,104,105,107,110] and high-quality studies[25,59,63,65,66,68,69]. In 14 studies, women with and without PCOS were matched for age[32,59,63,66,68,70–78] and in seven for both age and BMI[63,66,68,72,73,77,78]; PCOS was associated with increased odds of GDM in age-matched or age- and BMI-matched studies.

### Gestational hypertension
Forty-one studies reported gestational hypertension in 5978 women with PCOS and 32230 women without PCOS[17,18,20,22,32,41,49,56–63,65,66,68,69,71,74,76,78,82,83,88,92,94,95,97,98,102,103,105,107,109,111–115]. Two studies were excluded from the meta-analysis because of overlapping participants[62,109]. Gestational hypertension was defined as

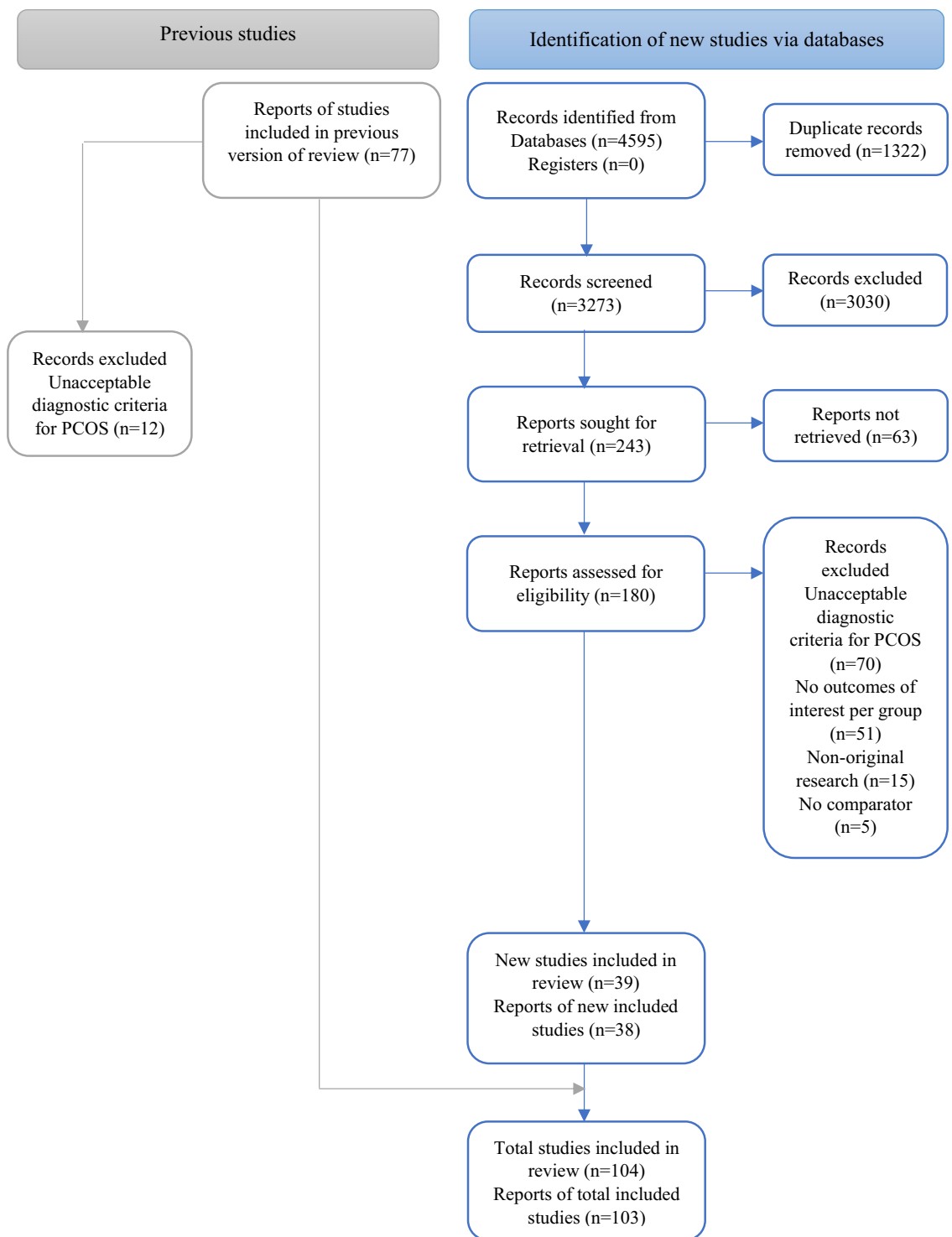

**Fig. 1 | PRISMA flow diagram of study selection.** This figure illustrates the PRISMA (Preferred Reporting Items for Systematic Reviews and Meta-Analyses) flow diagram detailing the study selection process. The diagram includes the number of records identified, screened, assessed for eligibility, and included in the meta-analysis.

systolic blood pressure ≥ 140 mmHg or diastolic blood pressure ≥ 90 mmHg in five studies[65,69,74,88,104], systolic blood pressure ≥ 140 mmHg or diastolic blood pressure ≥ 90 mmHg after 20 weeks of pregnancy in nine studies[18,59,66,76,78,81,83,103,113], systolic blood pressure ≥ 140 mmHg or diastolic blood pressure ≥ 90 mmHg at two occasions more than a day apart in one study[107], systolic blood pressure ≥ 140 mmHg or diastolic blood pressure ≥ 90 mmHg at two occasions more than six hours apart after 20 weeks of pregnancy in one study[114], systolic blood pressure ≥ 140 mmHg or diastolic blood pressure ≥ 90 mmHg at two occasions

more than six hours apart after 20 weeks of pregnancy and normalized by four to six weeks postpartum in one study[60], systolic blood pressure ≥ 140 mmHg or diastolic blood pressure ≥ 90 mmHg at two occasions after 20 weeks of pregnancy or systolic blood pressure ≥ 150 mmHg or diastolic blood pressure ≥ 100 mmHg during labor in two studies[32,71], systolic blood pressure ≥ 140 mmHg or diastolic blood pressure ≥ 90 mmHg without proteinuria after 20 weeks of pregnancy in two studies[18,41], systolic blood pressure ≥ 140 mmHg or diastolic blood pressure ≥ 90 mmHg with or without proteinuria after 20 weeks of

## Table 1 | Association of polycystic ovary syndrome with pregnancy outcomes

| Outcome | N. studies | Effect size [95% CI] | $I^2$ |
|---|---|---|---|
| Gestational weight gain (kg) | 16 | MD 0.96 [0.01, 1.90] | 88.1% |
| Post ART pregnancies | 1 | MD −1.10 [−2.53, 0.33] | .% |
| Prospective design | 6 | MD 1.30 [0.27, 2.33] | 82.3% |
| High quality | 3 | MD 3.24 [−0.72, 7.20] | 93.3% |
| Age[a] matched | 7 | MD 2.16 [−0.19, 4.50] | 90.8% |
| BMI[b] matched | 4 | MD 3.10 [−0.77, 6.98] | 93.1% |
| Miscarriage | 44 | OR 1.49 [1.20, 1.85] | 82.8% |
| Post ART pregnancies | 29 | OR 1.23 [1.02, 1.49] | 64.2% |
| Prospective design | 19 | OR 1.81 [1.22, 2.69] | 74.6% |
| High quality | 6 | OR 3.38 [2.03, 5.64] | 44.8% |
| Age matched | 7 | OR 3.62 [2.47, 5.29] | 0.0% |
| BMI matched | 6 | OR 3.63 [2.47, 5.32] | 0.0% |
| Gestational diabetes | 55 | OR 2.41 [1.95, 2.99] | 81.9% |
| Post ART pregnancies | 9 | OR 1.70 [1.03, 2.80] | 82.9% |
| Prospective design | 17 | OR 4.08 [2.61, 6.40] | 76.3% |
| High quality | 7 | OR 2.62 [1.13, 6.07] | 87.1% |
| Age matched | 14 | OR 2.05 [1.27, 3.31] | 52.4% |
| BMI matched | 7 | OR 2.85 [1.41, 5.78] | 60.5% |
| Gestational hypertension | 39 | OR 2.20 [1.82, 2.67] | 53.3% |
| Post ART pregnancies | 4 | OR 1.84 [1.18, 2.85] | 0.0% |
| Prospective design | 15 | OR 2.39 [1.72, 3.31] | 39.3% |
| High quality | 9 | OR 2.19 [1.32, 3.64] | 41.4% |
| Age matched | 12 | OR 2.54 [1.73, 3.73] | 0.0% |
| BMI matched | 7 | OR 2.41 [1.37, 4.26] | 8.5% |
| Pre-eclampsia | 34 | OR 2.30 [1.88, 2.82] | 28.1% |
| Post ART pregnancies | 2 | OR 3.06 [1.01, 9.25] | 0.0% |
| Prospective design | 14 | OR 2.73 [1.90, 3.92] | 41.0% |
| High quality | 6 | OR 3.05 [1.20, 7.80] | 51.3% |
| Age matched | 10 | OR 2.81 [1.47, 5.36] | 33.8% |
| BMI matched | 7 | OR 2.39 [1.14, 4.99] | 39.5% |
| Eclampsia | 1 | OR 1.16 [0.44, 3.08] | .% |
| Post ART pregnancies | 1 | OR 1.16 [0.44, 3.08] | .% |
| Prospective design | 0 | – | – |
| High quality | 0 | – | – |
| Age matched | 1 | OR 1.16 [0.44, 3.08] | .% |
| BMI matched | 0 | – | – |
| Induction of labor | 8 | OR 1.62 [0.97, 2.70] | 81.0% |
| Post ART pregnancies | 0 | – | – |
| Prospective design | 5 | OR 1.67 [1.10, 2.54] | 71.4% |
| High quality | 1 | OR 1.25 [0.56, 2.77] | .% |
| Age matched | 1 | OR 1.25 [0.56, 2.77] | .% |
| BMI matched | 1 | OR 1.25 [0.56, 2.77] | .% |
| Instrumental delivery | 10 | OR 1.18 [0.91, 1.53] | 0.0% |
| Post ART pregnancies | 0 | – | – |
| Prospective design | 6 | OR 1.26 [0.84, 1.88] | 10.0% |
| High quality | 3 | OR 0.84 [0.35, 2.04] | 13.2% |
| Age matched | 4 | OR 1.02 [0.58, 1.80] | 8.2% |
| BMI matched | 4 | OR 1.02 [0.58, 1.80] | 8.2% |
| Cesarean section | 37 | OR 1.23 [1.06, 1.43] | 63.5% |
| Post ART pregnancies | 7 | OR 0.96 [0.83, 1.12] | 3.9% |
| Prospective design | 15 | OR 1.44 [1.07, 1.95] | 73.5% |
| High quality | 7 | OR 1.58 [1.19, 2.09] | 0.0% |
| Age matched | 9 | OR 1.57 [1.24, 2.00] | 4.1% |

## Table 1 (continued) | Association of polycystic ovary syndrome with pregnancy outcomes

| Outcome | N. studies | Effect size [95% CI] | $I^2$ |
|---|---|---|---|
| BMI matched | 6 | OR 1.57 [1.19, 2.07] | 0.0% |
| Perinatal depression | 1 | 1.58 [0.87, 2.88] | .% |
| Post ART pregnancies | 0 | – | – |
| Prospective design | 0 | – | – |
| High quality | 0 | – | – |
| Age matched | 0 | – | – |
| BMI matched | 0 | – | – |

*ART* assisted reproductive technology, *BMI* body mass index, *MD* mean difference, *OR* odds ratio.
[a]Age is reported in years.
[b]BMI is reported in kg/m².

pregnancy in two studies[92,95], systolic blood pressure ≥ 140 mmHg or diastolic blood pressure ≥ 90 mmHg with or without proteinuria[68] in the third trimester in two studies[97], and diastolic blood pressure > 90 mmHg at two occasions over pregnancy in one study[98]. The remaining 22 studies did not provide a definition. The odds of gestational hypertension were higher in women with PCOS compared with women without PCOS (OR: 2.20; 95% CI: 1.81, 2.69). Sensitivity analysis showed that after exclusion of studies in which women were taking metformin after conception[17,18,20], gestational hypertension remained higher in PCOS (2.22; 1.83, 2.70). Cumulative meta-analysis suggested that the odds of gestational hypertension in women with and without PCOS did not substantially change over time; however, two studies published before 2001 contributed to heterogeneity. The higher odds of gestational hypertension were retained in post-ART pregnancies[32,41,49,95], prospective[56,58,60,61,63,65,66,76,78,83,88,94,102,105,107] and high-quality studies[56,59–61,63,65,66,68,69]. Three studies reported gestational hypertension in pregnancies with GDM[56–58]; the odds of gestational hypertension were higher in women with PCOS (2.57; 1.71, 3.88). In 12 studies, women with and without PCOS were matched for age[32,56,59–61,63,66,68,71,74,76,78] and in seven for both age and BMI[56,60,61,63,65,66,68,78]; PCOS was associated with increased odds of gestational hypertension in age-matched or age- and BMI-matched studies.

## Pre-eclampsia

Thirty-seven studies reported pre-eclampsia in 4999 women with PCOS and 24660 women without PCOS[14–16,20,21,32,41,55–58,60–63,66,69,71,74,77,78,81–83,88,94,97,98,102,103,105,107–109,112,113,116]. Three studies were excluded from the meta-analysis because of overlapping participants[15,62,109]. Pre-eclampsia was defined as gestational hypertension along with proteinuria in three studies[66,78,116], gestational hypertension along with proteinuria >2+ by urine stick in two studies[16,98], gestational hypertension along with proteinuria >300 mg in 24 h in thirteen studies[14,32,55,58,60,69,74,75,77,83,88,103,104], gestational hypertension along with proteinuria >300 mg in 24 h or ≥1+ by urine stick in one study[102], gestational hypertension along with proteinuria or end-organ dysfunction in two studies[41,113], gestational hypertension along with proteinuria >300 mg in 24 h or a spot urine protein/creatinine ratio of ≥30 mg/mmol or when the hemolysis elevated liver enzymes low platelets syndrome was present in one study[107], gestational hypertension along with proteinuria 100 mg/dL or >2+ in at least two random urine specimens collected six or more hours apart or >300 mg in 24 h and/or pathologic edema in three studies[71,81,115]. The remaining 22 studies did not provide a definition. The odds of pre-eclampsia were higher in women with PCOS compared with women without PCOS (OR: 2.30; 95% CI: 1.87, 2.82). Sensitivity analysis showed that, after exclusion of studies in which women were taking metformin after conception[16,20] or conceived after bariatric surgery[21], pre-eclampsia remained higher in PCOS (2.36; 1.93, 2.90). Cumulative meta-analysis

suggested a diminishing magnitude in the odds of pre-eclampsia in women with and without PCOS over time.

The higher odds of pre-eclampsia were retained in post-ART pregnancies[32,41] and prospective[56,58,60,61,63,66,78,83,88,94,102,105,107,116] and high-quality studies[56,60,61,63,66,69]. Four studies reported pre-eclampsia in pregnancies with GDM[55–58]; the odds of pre-eclampsia were higher in women with PCOS (2.72; 1.77, 4.19). In 10 studies, women with and without PCOS were matched for age[32,56,60,61,63,66,71,74,77,78] and in seven for both age and BMI[56,60,61,63,66,77,78]; PCOS was associated with increased odds of pre-eclampsia in age-matched or age- and BMI-matched studies.

### Eclampsia
Two studies reported eclampsia in 46 women with PCOS and 246 women without PCOS[32,60]. One study reported no pregnancies affected by eclampsia[60]; therefore, only one study[32] in post-ART pregnancies was used in the meta-analysis for generating the OR. The odds of eclampsia were similar in women with and without PCOS (OR: 1.16; 95% CI: 0.44, 3.08). Participants in this study were not taking metformin after conception.

### Induction of labor
Eight studies reported the induction of labor in 1029 women with PCOS and 10052 women without PCOS[14,21,56,82,102,104,105,107]. The odds of induction of labor were similar in women with and without PCOS (OR: 1.62; 95% CI: 0.97, 2.70). Sensitivity analysis showed that after exclusion of a study in which women conceived after bariatric surgery[21], the induction of labor was higher in the PCOS group (1.69; 1.01, 2.85). Cumulative meta-analysis suggested that the odds of induction of labor in women with and without PCOS did not substantially change over time; however, one study published before 2012 contributed to heterogeneity. No studies reported the induction of labor in post-ART pregnancies. The odds of induction of labor were higher in prospective studies[56,102,104,105,107] but similar in a single high-quality, age- and BMI-matched study that reported outcomes in pregnancies with GDM[56].

### Instrumental delivery
Ten studies reported instrumental delivery in 1517 women with PCOS and 8570 women without PCOS[21,56,62,66,72,98,103,105,111,116]. The odds of instrumental delivery were similar in women with and without PCOS (OR: 1.18; 95% CI: 0.91, 1.53). Sensitivity analysis showed that after exclusion of a study in which women conceived after bariatric surgery[21], instrumental delivery remained similar in women with and without PCOS (1.17; 0.90, 1.52). Cumulative meta-analysis suggested that the odds of instrumental delivery in women with and without PCOS did not substantially change over time. The odds of instrumental delivery were similar in prospective[56,62,66,105,116] and high-quality studies[56,66,88]. No studies reported instrumental delivery in post-ART pregnancies. Only one study reported instrumental delivery in pregnancies with GDM[56], whereby the odds were similar in women with and without PCOS (1.35; 0.22, 8.41). In four studies, women with and without PCOS were matched for both age and BMI[56,66,72,88], in which PCOS was not associated with instrumental delivery.

### Cesarean section
Thirty-seven studies reported cesarean section in 3590 women with PCOS and 23017 women without PCOS[14,17,21,25,33,38,41,44,45,53,56–59,62,63,66,68,71,74,78,82,85,86,94,95,97,98,102–107,111,116,117]. The odds of cesarean section were higher in women with PCOS compared with women without PCOS (OR: 1.23; 95% CI: 1.06, 1.43). Sensitivity analysis showed that after exclusion of studies in which women were taking metformin after conception[17] or conceived after bariatric surgery[21], cesarean section remained higher in PCOS (1.23; 1.06, 1.44). Convergence could not be achieved during tau$^2$ estimation in random-effects cumulative meta-analysis. The higher odds of cesarean section were retained in prospective[38,56,58,62,63,66,78,85,94,102,104,105,107,116,117] and high-

quality studies[25,56,59,62,63,66,68] but were not retained for post-ART pregnancies[25,33,38,40,41,44,53]. Three studies reported cesarean section in pregnancies with GDM[56–58], the odds of which were higher in women with PCOS (1.69; 1.19, 2.39). In nine studies, women with and without PCOS were matched for age[56,59,62,63,66,68,71,74,78] and in six for both age and BMI[56,62,63,66,68,78]; PCOS was associated with increased odds of cesarean section in age-matched or age- and BMI-matched studies.

### Perinatal depression
One study reported perinatal depression in 3590 women with PCOS and 23017 women without PCOS[118], with similar odds between groups (OR: 1.58; 95% CI: 0.87, 2.88). Participants in this study were not taking metformin after conception.

### Meta-regression
Significant heterogeneity ($I^2 > 50\%$) was observed for studies reporting miscarriage, GDM, gestational hypertension, gestational weight gain, induction of labor and cesarean section. There was an insufficient number of studies (<10) for the induction of labor to perform meta-regression.

Where reported (Table 2), age was lower in women with PCOS for miscarriage and GDM. However, BMI was higher in women with PCOS for miscarriage, GDM, gestational hypertension, gestational weight gain and cesarean section. In univariate meta-regression, variation in age was associated with increased odds of miscarriage, with a one-year increase in age being associated with 32% higher odds of miscarriage. Age reduced the tau$^2$ value for miscarriage from 0.31 to 0.22, indicating that 29% of between-study variance may be attributable to age. However, variation in BMI across studies was not associated with increased odds of any of the outcomes in the meta-regression.

## Discussion
In this comprehensive systematic review, meta-analysis, and meta-regression of 104 published articles in 17384 women with and 89306 women without PCOS, women with PCOS were younger, had higher BMI around conception and had higher gestational weight gain. They were more likely to experience pregnancy complications, including miscarriage, GDM, hypertension, pre-eclampsia, induction of labor, and cesarean section. Induction of labor became higher in PCOS in the sensitivity analysis, excluding women who conceived after bariatric surgery and in prospective studies. In sensitivity meta-analyses, PCOS remained associated with higher miscarriage, GDM, hypertension and pre-eclampsia in post-ART pregnancies, prospective and high-quality studies and age-matched and age- and BMI-matched studies. Cesarean section remained associated with PCOS in prospective and high-quality studies and age-matched and age- and BMI-matched studies, but it was not associated with PCOS in post-ART pregnancies.

Overweight and obesity affect approximately 70% of women in the United States and are increasing worldwide[119]. There is also a global increase in gestational weight gain above recommendations[120], which is a key independent contributor to obesity in the female population. Suboptimal weight around conception and insufficient or excessive gestational weight gain are associated with adverse maternal and off-spring outcomes[121]. Aging is associated with a higher weight and wor-sened pregnancy outcomes[122]. We found that women with PCOS conceive at a higher BMI compared with women without PCOS and have higher gestational weight gain. These were assessed only and for the first time in our previous systematic review and meta-analysis[8] and have been reaffirmed in these updated analyses. Mean differences in gestational weight gain did not have substantial variation over time. Women with PCOS were also younger. We were underpowered for the majority of the sensitivity analyses by age-matched or age- and BMI-matched studies. Our findings suggest that the higher gestational weight gain in PCOS is likely related to factors other than PCOS status, such as age and BMI. However, the higher odds of miscarriage, GDM,

**Table 2 | Effect of age and BMI on the association of polycystic ovary syndrome with pregnancy outcomes**

| Outcomes | Age (year) | | | BMI (kg/m²) | | |
|---|---|---|---|---|---|---|
| | N | Effect size | Tau² | N | Effect size | Tau² |
| **Gestational weight gain (kg)** | | | | | | |
| MD [95%CI] | 15 | −0.47 [−0.97, 0.02] | – | 15 | 2.24 [1.02, 3.46] | – |
| Univariate coefficient (95% CI) | 15 | −0.71 [−2.09, 0.66] | 0.30 | 15 | 0.02 [−0.22, 0.27] | 0.33 |
| **Miscarriage** | | | | | | |
| MD [95%CI] | 25 | −1.02 [−1.45, −0.60] | – | 23 | 1.63 [1.17, 2.09] | – |
| Univariate coefficient (95% CI) | 25 | 0.32 [0.13, 0.51] | 0.22 | 23 | −0.05 [−0.22, 0.12] | 0.42 |
| **Gestational diabetes** | | | | | | |
| MD [95%CI] | 34 | −0.41 [−0.77, −0.04] | – | 29 | 1.75 [1.23, 2.27] | – |
| Univariate coefficient (95% CI) | 34 | 0.02 [−0.27, 0.31] | 0.46 | 29 | 0.08 [−0.13, 0.30] | 0.40 |
| **Gestational hypertension** | | | | | | |
| MD [95%CI] | 25 | −0.25 [−0.64, 0.14] | – | 22 | 1.58 [0.88, 2.27] | – |
| Univariate coefficient (95% CI) | 25 | −0.03 [−0.26, 0.20] | 0.06 | 22 | 0.06 [−0.13, 0.24] | 0.12 |
| **Cesarean section** | | | | | | |
| MD [95%CI] | 22 | −0.36 [−0.74, 0.02] | – | 19 | 1.49 [1.01, 1.96] | – |
| Univariate coefficient (95% CI) | 22 | 0.06 [−0.15, 0.28] | 0.08 | 19 | 0.07 [−0.03, 0.16] | 0.04 |

*BMI* body mass index, *MD* mean difference.

gestational hypertension, pre-eclampsia, and cesarean section are probably independent of age and BMI. The 2023 International Evidence-based PCOS Guideline[2] recommends early lifestyle intervention in women with PCOS to manage weight and improve fertility and pregnancy outcomes. However, the association of PCOS with increased odds of miscarriage, GDM, gestational hypertension, pre-eclampsia and cesarean section is probably independent of age and/or BMI.

Miscarriage affects 15.3% of confirmed pregnancies worldwide[123]. Here, we found that women with PCOS have 49-53% higher odds of miscarriage, and these increased odds were independent of time and ART and were retained in both prospective and high-quality studies. Age was a risk factor for increasing the odds of miscarriage independent of PCOS in univariate meta-regression. Our findings of increased odds of miscarriage in PCOS and the association with age are consistent with prior literature[8] and are not reported in other systematic reviews[9–12]. Here, we report for the first time that higher odds of miscarriage were observed in a younger population of women with PCOS. While ovulation rates improve in women with PCOS as they age, successful pregnancy and live births decrease with age[124]. This highlights that the odds of miscarriage would probably be even higher in PCOS if age was similar between the two groups. The magnitude for higher odds of miscarriage in PCOS decreased in post-ART pregnancies, but inconsistent with our original meta-analysis[8], it remained higher in PCOS, with significant heterogeneity. However, the recent findings reported herein are based on a larger number of included studies and more accurate PCOS diagnosis and thus are likely more reliable.

Gestational diabetes affects up to 30% of pregnancies, depending on diagnostic criteria and ethnicity[125]. It is associated with increased rates of cesarean section, preterm labor, macrosomia, and large-for-gestational age offspring[126] and with increased maternal risk for cardiovascular and cerebrovascular diseases later in life[127]. We found that PCOS is associated with 2-3-fold higher odds of GDM, and the increased odds were independent of time and ART and were retained in both prospective and high-quality studies. Where information was provided, women with PCOS were younger and had a higher BMI than women without PCOS. In those with PCOS, neither age nor BMI was associated with increased odds of GDM in the adjusted analysis, highlighting that the higher odds of GDM in PCOS are likely due to factors other than increased BMI. The magnitude for higher odds of GDM in PCOS decreased in post-ART pregnancies, but inconsistent

with our prior meta-analysis[8], it remained higher in PCOS. The attenuated odds are probably due to BMI restrictions for ART and poorer treatment outcomes in individuals with higher BMI[128]. Compared to the previous review, however, the validity of findings in this review is increased by the larger number of included studies and more accurate PCOS diagnosis. The 2023 International Evidence-based PCOS Guideline recommends that women with PCOS are offered an oral glucose tolerance test (OGTT) for early diagnosis and intervention, ideally at preconception; if not performed at preconception, at the first antenatal visit and then repeated at 24-28 weeks of pregnancy.

Hypertensive disorders of pregnancy affect 8.5-16.7% of pregnancies, depending on ethnicity[129]. Gestational hypertension and pre-eclampsia are leading causes of pregnancy-related maternal mortality and are associated with 2-3-fold increases in long-term mortality from maternal cardiovascular disease[130]. We found that PCOS is associated with 2-3-fold higher odds of hypertensive disorders of pregnancy, independent of ART, and this was retained in both prospective and high-quality studies. Over time, the odds of pre-eclampsia diminished but remained higher in PCOS. This could be due to later studies including younger women with PCOS[131] who are less likely to have chronic hypertension[132]. Where information was provided, women with PCOS were of similar age as women without PCOS and had higher BMI, with neither being associated with increased odds of gestational hypertension on meta-regression. This highlights that the higher odds of hypertensive disorders of pregnancy in PCOS were due to factors other than increased BMI. We confirm the findings of our previous meta-analysis[8], which showed that the higher odds of hypertensive disorders of pregnancy are likely independent of, but worsened by, GDM; however, we were underpowered for this sensitivity analysis. The higher odds of hypertensive disorders of pregnancy in PCOS were retained in post-ART pregnancies. This is inconsistent with the findings of our previous meta-analysis[8], although the results are likely more reliable in the present review given the larger number of studies included here. Based on these findings, the 2023 International Evidence-based PCOS Guideline[2] has recommended monitoring of blood pressure in women with PCOS, ideally commencing in the pre-conception period.

Although spontaneous vaginal delivery is the most common type of delivery, the rates of induction of labor, instrumental delivery and cesarean section are increasing[133]. Induced and/or operative deliveries are indicated in high-risk pregnancies for mothers or offspring. The

underlying conditions and the interventions may both increase the risk of hemorrhage, infection, and rehospitalization[134]. We found that women with PCOS have higher odds of cesarean section, which were retained in both prospective and high-quality studies. However, the odds were similar in post-ART pregnancies in women with and without PCOS. For the induction of labor and instrumental delivery, the odds were similar in women with and without PCOS, which may have been masked by the increased odds of cesarean section in women with PCOS. Our findings are consistent with the findings of a prior systematic review[8]. While the number of included studies was small, the higher odds of cesarean section appear to be independent of but worsened by GDM, which is consistent with our previous meta-analysis[8].

The strengths of this systematic review and meta-analysis are the inclusion of the largest number of observational studies and participants across five continents to date. We excluded less reliable diagnostic methods such as self-reported PCOS status and ICD codes, as these can introduce inaccuracies due to recall bias or misinterpretation of the diagnosis and the inherent variability in diagnostic criteria across medical organizations[135,136]. Coupled with the heterogeneous nature of PCOS presentations, these less reliable methods can increase the potential for misdiagnosis[137,138], thus capturing those with more severe symptoms and/or those not meeting the diagnostic criteria and distorting the findings on pregnancy outcomes. For the first time, we reported the mean age and BMI corresponding to each outcome of interest and performed cumulative analyses to assess the potential impact of time on the outcomes. We performed sensitivity analyses to assess the potential impact of confounders, and we also performed meta-regression to assess the impact of age and BMI where substantial heterogeneity was observed, a pioneering approach from our previous systematic review. The certainty of evidence was assessed for individual outcomes using the GRADE approach. A limitation of this review is the inclusion of studies that were published in English, which contributes to potential language bias since English studies are more likely to demonstrate positive findings. Other limitations include the unclear or inconsistent definitions for some of the outcomes; small number of studies for sensitivity analyses by age-matched, BMI-matched and high-quality studies; and the insufficient number of studies for meta-analyses on eclampsia and perinatal depression. Furthermore, 64.1% of the included studies were of moderate to high risk of bias mainly due to high confounding bias (57.7%) followed by high selection bias (31.7%); however, we conducted sensitivity analyses of high-quality studies that confirmed the overall findings for the majority of outcomes. As this was a meta-analysis of aggregate-level data, we could only explore confounding factors in relation to age, BMI, or post-ART populations; hence, other potentially important confounders could not be accounted for.

In this systematic review and meta-analysis, we found that women with PCOS are more likely to experience pregnancy complications, which are independent of but likely exacerbated by increased age and BMI and by assisted conception. The increased odds for pregnancy complications were impacted by the quality of the studies. Women with PCOS are more likely to have pregnancy-related obesity, which could be both a risk factor for and a consequence of metabolic disorders in pregnancy. This systematic review and meta-analysis of 106690 pregnancies in women with and without PCOS directly informs the 2023 International Evidence-based PCOS Guideline recommendation to consider PCOS as a risk factor for pregnancy complications in preconception and in antenatal care guidelines to enable timely screening, proper monitoring, and early intervention. Additional risk factors, such as age, BMI and ART, should be considered in women with PCOS. Individual patient data meta-analyses with sufficient statistical power are warranted to further investigate the impact of PCOS features and to clarify important confounders and sources of heterogeneity in the relationship between PCOS and pregnancy outcomes.

## Methods

### Search strategy and selection criteria

This systematic review and meta-analysis is an update of previously published systematic reviews[8,139] and was conducted in accordance with the Meta-Analyses and Systematic Reviews of Observational Studies (MOOSE) guidelines[140]. Given that the majority of studies on pregnancy outcomes in prospectively identified PCOS come from selected populations with a high-risk profile[141], we did not prioritize the distinction between retrospective and prospective study designs; rather, our emphasis was on the overall quality of the studies. The protocol for the original review was prospectively registered in PROSPERO (CRD 42017067147). The methods and results of the previous search with the inclusion of publications up to the 4th of April 2017 have been previously published[8,139]. The search was limited to English language studies and updated (MBK) from 2017 to 13th of July 2022 through Medline, Medline in-process, and other non-indexed citations, EMBASE, and all EBM reviews, including Cochrane Database of Systematic Reviews, Cochrane Clinical Answers, Cochrane Central Register of Controlled Trials, American College of Physicians Journal Club, Cochrane Methodology Register, Health Technology Assessments, The Database of Abstracts of Reviews of Effectiveness and the National Health Service Economic Evaluation Database. Bibliographies of recent and relevant systematic reviews and meta-analyses were searched to identify additional studies. As was done previously, search terms included a broader number of outcomes, distributed to two systematic reviews and meta-analyses[8,139].

Eligible studies included studies reporting observational data on gestational weight gain, miscarriage, GDM, gestational hypertension, pre-eclampsia, eclampsia, induction of labor, instrumental delivery, cesarean section, and perinatal depression in women with and without PCOS. In a protocol amendment, we only included studies in which the criteria used for PCOS diagnosis fulfilled the Rotterdam criteria[136]. Pregnancy outcomes were accepted according to the primary studies' definitions.

Studies published in languages other than English; case reports, case series, editorials, scoping and narrative reviews; studies using self-reported or International Classification of Diseases (ICD) for PCOS diagnosis; and studies not reporting the outcomes of interest in the two groups of PCOS versus non-PCOS were excluded.

Two reviewers (MBK, SH, or SS) independently screened studies by titles and abstracts and reviewed full texts for eligibility. Discrepancies were discussed and resolved through consensus or arbitration between reviewers. Studies identified through the new search were combined with those published before April 2017. Studies not meeting the new criteria needed for PCOS diagnosis were excluded (i.e., studies captured in the 2017 reviews[8,139] that used self-report or ICD for PCOS diagnosis were excluded from the current review).

### Data analysis

Data extraction and quality appraisal were independently performed by the same reviewers. Using an a priori researcher-developed data extraction form, data were extracted from each study. These included the author, year of publication, study design, study location, participant characteristics, and frequency/mean and standard deviation of outcomes per group. Participant characteristics included age, BMI, mode of conception, existing medical conditions, and medications used during pregnancy.

When participants were overlapping between multiple publications for the same outcome, data from the study with the largest sample size were included in the meta-analysis.

the risk of bias was independently assessed by two reviewers (MBK, SH, or SS) using the Newcastle-Ottawa Scale (NOS) for non-randomized studies for selection, comparability, and outcome ascertainment[142]. Studies were considered high-quality if they scored at least three stars in selection, one star in comparability, and two stars

in outcome ascertainment. Studies with two stars in selection, a minimum of one star in comparability, and two stars in outcome ascertainment were considered fair quality. Studies meeting neither of these two thresholds were considered low quality.

We performed random-effects meta-analyses to generate pooled effect estimates, reported as the mean differences (MDs) or odds ratios (ORs) with 95% confidence intervals (CIs), for the association between PCOS status and pregnancy outcomes. Between-study heterogeneity was assessed using the $I^2$ statistic; $I^2$ above 50% was deemed substantial heterogeneity[143]. Sensitivity analyses were performed, excluding studies in which participants were taking metformin after conception or conceived after bariatric surgery. Cumulative random-effects meta-analyses based on Hedge's adjusted g and its 95% confidence intervals (95% CI)[144] were performed to explore the effect of time (publication year) on the association of PCOS with each outcome of interest. Publication bias was assessed using funnel plots. We assessed the certainty of evidence for each outcome according to the Grading of Recommendations, Assessment, Development and Evaluations (GRADE) system[145] using Gradepro software[146].

Sensitivity analyses of the outcomes by conception with ART (in vitro fertilization, in vitro maturation intracytoplasmic sperm injection, zygote intrafallopian transfer, and gamete intro-fallopian transfer) and prospective and high-quality studies as well as by age- and BMI-matched designs were also performed after exclusion of studies in which women were taking metformin after conception or conceived after bariatric surgery.

Restricted maximum likelihood (REML)-based random effects meta-regression was performed to explore the effects of variations in age and BMI on each outcome of interest if there were equal to or greater than 10 studies per coefficient. Effect sizes of age and BMI for individual studies were used in meta-regression analyses. The percentage of between-study variance explained by the model (tau²) was estimated using the Knapp–Hartung modification. Normal distributions for mean values were checked using skewness-kurtosis tests. As age and BMI were not both significant ($p < 0.01$) for any of the outcomes on univariate meta-regression analyses, multivariable meta-regression could not be performed. Statistical significance was defined as two-sided $P < 0.05$. All statistical analyses were performed using Stata version 17 (StataCorp, 14 College Station, Texas, USA).

### Reporting summary

Further information on research design is available in the Nature Portfolio Reporting Summary linked to this article.

## Data availability

Data that support the findings of this study have been deposited in https://data.mendeley.com/datasets/npy96r94p2/1 (https://doi.org/10.17632/npy96r94p2.1). Source data are provided with this paper.

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

## Acknowledgements

Eszter Vanky and Aya Mousa contributed equally as joint senior authors. We extend our gratitude to the colleagues who contributed to the initial systematic review published in 2019. While these collaborators were not directly involved in this update, their contributions to the foundational work have been instrumental in shaping the direction and scope of the current study. This work is supported by the Australian National Health and Medical Research Council (NHMRC) funded Centre for Research Excellence in Women's Health in Reproductive Life (CRE-WHiRL) [APP#1171592] (HJT). The funder of the study had no role in study design, data extraction, data analysis, data interpretation, writing of the manuscript, or the decision to submit the manuscript for publication.

## Author contributions

M.B.K. contributed to the concept and design and performed the systematic search and statistical analysis. M.B.K., S.S., and S.H. performed screening, data extraction, quality appraisal, and drafting of the paper. H.J.T. obtained funding and led the overarching guideline process. C.T.T. and A.M. were evidence synthesis leads for the guideline including this study. E.V. was the clinical expert lead in the guideline committee. M.B.K., S.S., S.H., C.H.L., T.T.P., D.R., C.T.T., H.J.T., E.V., and A.M. contributed to the concept and design and provided substantial contributions to the drafting of the work, including critical revision for important intellectual content. M.B.K., S.S., S.H., C.H.L., T.T.P., D.R., C.T.T., H.J.T., E.V., and A.M. had full access to the data, were responsible for study integrity and approved the final version of the manuscript for publication.

## Competing interests

T.P. is supported by Novo Nordisk and Sigrid Jusélius Foundation. EV is supported by Novo Nordisk and Merck as a lecturer and advisor for clinical studies. H.J.T. and A.M. are supported by NHMRC fellowships [APP#2009326 and APP#1161871, respectively]. The remaining authors declare no competing interests.
