## [Peer Review File · Nature Communications]

REVIEWERS' COMMENTS

Reviewer #1 (Remarks to the Author):

From the perspective of this referee, the current meta analysis provides insufficient novel insight (on top of much knowlegde generated by many previous publications on this particular topic), to justify publication in the present journal.

Publication in a journal focussing on reproduction/ fertility seems more appropriate.

Reviewer #4 (Remarks to the Author):

I have reviewed the manuscript before. The authors answered my concerns. I have nothing further.

Reviewer #5 (Remarks to the Author):

This is a very well conducted Systematic Review and it is a very good and necessary update regarding pregnancy complications in PCOS patients.

In my opinion the concerns raised by the other reviewers have been properly addressed .

One important issue that could be addressed in the discussion is that the genetic and environmental differences between the populations represented in the studies might change the prevalence of the pregnancy outcomes. For instance, in Latin American countries the incidence of insulin resistance is higher than in European countries and this might change the prevalence of Gestational Diabetes in the PCOS population from this paricular area. As studies from different areas (Europe, America, Asia, etc) were included, it would be interesting to show a comparison of the prevalences of

pregnancy outcomes between different geographic areas. This would aid in the applicability of this data in countries with a different genetic and cultural background.

I have no further questions to the authors.

My congratulations for a very good work.

Re: submission of a revised original research manuscript entitled “Meta-Analysis of Pregnancy Outcomes in women with Polycystic Ovary Syndrome for the International Guideline”.

We thank the editor and reviewers for the constructive feedback. We have amended our manuscript accordingly, with point-by-point responses to the comments outlined below.

REVIEWERS' COMMENTS

Reviewer #1 (Remarks to the Author):

From the perspective of this referee, the current meta-analysis provides insufficient novel insight (on top of much knowledge generated by many previous publications on this particular topic), to justify publication in the present journal. Publication in a journal focussing on reproduction/ fertility seems more appropriate.

Response: The strengths of this study, compared to previous meta-analyses, were highlighted in the 'Strengths and Limitations' section of our manuscript. Unlike prior meta-analyses that focused solely on the odds of adverse pregnancy outcomes, our study additionally explores the impact of potential confounders such as age, BMI, and timing. This allows us to isolate the specific impact of PCOS on the outcomes for the first time. Our understanding is that the editors have already accepted the manuscript.

Reviewer #4 (Remarks to the Author):

I have reviewed the manuscript before. The authors answered my concerns. I have nothing further.

Response: Thanks.

Reviewer #5 (Remarks to the Author):

This is a very well conducted Systematic Review and it is a very good and necessary update

regarding pregnancy complications in PCOS patients.

In my opinion the concerns raised by the other reviewers have been properly addressed.

One important issue that could be addressed in the discussion is that the genetic and environmental differences between the populations represented in the studies might change the prevalence of the pregnancy outcomes. For instance, in Latin American countries the incidence of insulin resistance is higher than in European countries and this might change the prevalence of Gestational Diabetes in the PCOS population from this particular area. As studies from different areas (Europe, America, Asia, etc) were included, it would be interesting to show a comparison of the prevalences of pregnancy outcomes between different geographic areas. This would aid in the applicability of this data in countries with a different genetic and cultural background.

Response: Thank you for the reviewer's comment. We acknowledge the challenge of extracting data on participants' ethnicity due to limited or non-existent reporting. Analysis has to be confined to the countries where the studies were conducted, which serves as a surrogate but does not equate to ethnicity. Addressing this topic is beyond the scope of the current paper. We have work underway for self-identified ethnicity and country of birth using individual patient data analysis, which will address this gap in future.

I have no further questions to the authors.
My congratulations for a very good work.

Response: Thanks.